

# Analysis of factors influencing the pathological complete remission (ipCR) in patients with internal mammary lymph node metastasis after neoadjuvant chemotherapy

Yang Li and Yang Fei

Department of General Surgery, The First Medical Center of Chinese PLA General Hospital, Beijing, China

## ABSTRACT

**Objective**. To investigate the factors impacting pathological complete remission (ipCR) of the internal mammary lymph nodes in patients with internal mammary lymph node metastasis (IMLN) after adjuvant chemotherapy.

**Methods**. Sixty-five cases of primary breast cancer (BC) with IMLN metastasis who had received neoadjuvant chemotherapy (NAC) were retrospectively analyzed. Postoperative pathology was used to divide the patients into ipCR and non-ipCR groups. Univariate and multivariate analyses were performed on ipCR after NAC. A receiver operating characteristic (ROC) curve was used to evaluate the predictive value of the factors related to ipCR and a Kaplan-Meier curve was used to analyze prognosis.

**Results**. Twenty-nine (44.62%) of the 65 female patients received ipCR after NAC. Significant differences in hormone receptor (HR) negative and axillary pathological complete response (apCR) rates between the ipCR and non-ipCR group ($P < 0.05$). Multivariate logistic regression analysis showed that HR (OR = 2.698) and apCR (OR = 4.546) were the most significant factors that influenced ipCR ($P < 0.05$). The ROC curves showed that the area under the curves (AUC) for HR and apCR for the prediction of ipCR were 0.744 and 0.735 respectively. The AUC for the combined detection was 0.905. The average disease free survival (DFS) for patients in the ipCR group was 94.0 months which was significantly longer compared to patients in the non-ipCR group (64.2 months) ($\chi 2 = 4.265$, $P = 0.039$). No significant difference in OS was detected between the two groups ($P > 0.05$).

**Conclusions**. ipCR after NAC is correlated with HR and apCR. HR combined with apCR has value in predicting ipCR. ipCR has prognostic value in patients with IMLN metastasis and may have the potential to inform clinical decision-making. Further validation of these findings is required through larger-scale prospective studies.

# INTRODUCTION

Breast cancer (BC) is the most common malignancy in women and has major health impacts around the world. According to the latest global cancer data released by the World

Corresponding author
Yang Fei, feiyang0853@163.com

Health Organization International Agency for Research on Cancer, the number of new BC patients in 2020 reached 2.26 million and officially replaced lung cancer as the world's most common cancer (*Plichta, 2019*). In China, the incidence and mortality of BC continue to increase. The development of precision medicine has led to the molecular classification of BC subtypes towards the application of personalized treatments. BC can be divided into four molecular subtypes, specifically, luminal A, luminal B, HER-2 overexpression and triple-negative breast cancer (TNBC). BC treatment requires comprehensive interventions including surgery, radiotherapy and systemic treatments such as chemotherapy and targeted therapies.

The internal mammary lymph nodes (IMLNs) are an important lymph drainage system in breast tissue that has a prominent role in the metastasis of BC cells second only to the axillary lymph node (ALN). Lymph node metastasis is an important factor in determining the clinicopathological stage and in the prognosis and adjuvant treatment of BC patients. The internal mammary lymph nodes are an important part of the lymphatic drainage of BC as 10–40% of BC patients develop internal mammary lymph node metastasis. The IMLNs are the second most common site of lymph node metastasis in BC (*Zhang et al., 2015*). Lymph node metastasis in the inner mammary region is a factor in the staging of BC and an independent prognostic indicator (*Chen et al., 2008*).

The indications for biopsy or resection of internal mammary lymph nodes remain controversial due to a lack of improvement in prognosis and bleeding complications. In BC cases with IMLN metastasis, multimodal treatment combining breast surgery, internal mammary lymph node radiotherapy and systemic therapy is advocated (*Kim et al., 2021*). Neoadjuvant chemotherapy (NAC) is the standard and preferred treatment option for locally advanced and stage II~III; human epidermal growth factor receptor 2 (HER2)—positive and TNBC patients (*Akram et al., 2017*). NAC, also known as preoperative chemotherapy or induction chemotherapy, is widely used in clinical practice. It can achieve clinical downstaging and grading of tumors by reducing tumor mass, controlling tumor dissemination, and can increase the rate of breast conservation after surgery. The emergence of NAC has greatly improved the prognosis of BC patients. and is the main systemic treatment option. NAC can be used to convert 40–75% of cases with positive axillary lymph nodes into negative cases. However, some forms of BC are resistant to chemotherapy leading to delays in surgery. Studies have suggested the ALN status of BC patients after NAC indicates a pathological complete response (pCR) so ALN dissection is not required and avoids related complications.

pCR is an independent prognostic factor, independent of the intrinsic subtype of BC after neoadjuvant chemotherapy (*Bonnefoi et al., 2014*). *Mougalian et al. (2016)* demonstrated that patients with apCR after NAC have better long-term clinical outcomes compared to patients with residual axillary disease. However, few studies have analyzed the factors influencing ipCR after NAC. This article aims to study the factors of pathological complete remission of BC patients with IMLN metastasis after NAC and to provide a basis for individualized BC treatments.

## MATERIALS & METHODS

### Materials

A total of 65 patients who received NAC with lymph node metastasis in the inner mammary region in the breast center of our hospital from October 1, 2012, to October 1, 2022, were collected. All samples obtained in this study were approved by the ethics committee of the First Medical Center of Chinese PLA General Hospital (NO. 2023KY032-KS001). This study was conducted according to the ethical guidelines of the Declaration of Helsinki. The local ethics committee agreed to waive informed consent. The ipCR was defined as the absence of cancer metastasis in lymph nodes after modified extended radical mastectomy or IMLN dissection.

The patients were divided into an ipCR group (29 cases) and a non-ipCR group (36 cases) according to the pathological condition of the internal mammary region after surgery. The inclusion criteria were (1) BC patients diagnosed by histopathology, (2) ipsilateral IMLN metastasis confirmed by histopathology, (3) IMLN node dissection or modified extended radical surgery after NAC, (4) cases with complete clinical, pathology and follow-up data, and (5) patients who had been treated with at least two cycles of NAC. The exclusion criteria were cases of bilateral breast cancer, distant metastasis, and patients with a history of malignant disease.

### Methods

All patients received 3–8 cycles of NAC before surgery to reduce the volume of the tumor before surgery. Patients in the ipCR and non-ipCR groups were treated with taxanes or anthracycline-containing regimens. Specifically, the regimens were (1) AT: Doxorubicin 60 mg/m$^2$ iv drip d1 + docetaxel 75 mg/m$^2$ iv drip d1, 21 days as a cycle; (2) ET: Epirubicin 75 mg/m$^2$ iv drip d1 + docetaxel 75 mg/m$^2$ iv drip d1, 21 days as a cycle; (3) TAC: Docetaxel 75 mg/m$^2$ iv drip dl + doxorubicin 50 mg/m$^2$ iv drip dl + cyclophosphamide 500 mg/m$^2$ iv drip dl, 21 days for a cycle; (4) TEC: Docetaxel 75 mg/m$^2$ iv drip d1 + epirubicin 75 mg/m$^2$ iv drip d1 + cyclophosphamide 500 mg/m$^2$ iv drip d1, 21 days for a cycle; (5) TCbH: docetaxel 75 mg/m$^2$ iv drip dl, carboplatin AUC6 iv drip dl, trastuzumab 8 mg/kg first dose, followed by 6 mg/kg iv drip dl, 21 days as a cycle.

After completion of NAC, all patients required IMLN resection or modified extended radical mastectomy. The purpose of this surgery is to remove lymph nodes that may be affected by metastasis and to prevent further spread of cancer. After surgery, all patients received radiotherapy individualized according to the patient-specific characteristics.

Radiotherapy regimens were selected according to clinical stage, lesion size and lymph node infiltration. Patients were treated with six MV photons, once a day, five times a week. Patients were treated in the supine position using a breast bracket and the radiation field included the chest wall and supraclavicular lymph area. Three-dimensional conformal radiotherapy (3DCRT) was used in some cases with a total dose of 50 Gy delivered in 25 fractions of 2 Gy. Intensity-modulated radiotherapy (IMRT) was used in some cases with a total dose of 50 Gy delivered in 25 fractions of 2 Gy, or a total dose of 60.2 Gy, delivered in 28 fractions of 2.15 Gy. For volume-modulated arc therapy (VMAT) cases, a total dose

of 50 Gy was delivered in 25 fractions of 2 Gy or a total dose of 60.2 Gy was delivered in 28 fractions of 2.15 Gy.

Patients with HR-positive disease were treated with adjuvant endocrine therapy that included hormone receptor modulators (such as tamoxifen or cyclophosphamide) or selective estrogen receptor degrading agents (SERD). These drugs can prevent cancer recurrence by inhibiting or blocking the effect of estrogen on cancer cells. It should be emphasized that the specific treatment plan varied according to the condition of the patient and the pathological characteristics.

All selected cases were followed up by telephone. The deadline for follow-up was May 1st, 2023, and the median follow-up time of 38.5 months. The follow-up data included (1) whether the patient continued to receive endocrine therapy and molecular targeted therapy after discharge, (2) if the patient was regularly reviewed, had the tumor progressed, the time for progression, specific diagnosis and treatment after progression, and (3) survival of the patient and the specific cause of death in patients who had died.

Local and regional lymph node recurrence, distant metastasis and survival were recorded during follow-up. Disease-free survival (DFS) was defined as the time from surgery to tumor recurrence, death or the follow-up endpoint. Overall survival (OS) was defined as the time from surgery to death or the follow-up endpoint. DFS and OS were calculated from the date of the first operation. The endpoint of DFS was defined as local recurrence or distant metastasis. Local recurrence was defined as recurrence in the chest wall, axilla, supraclavicular region, IMLNs and soft tissues of the whole body and was confirmed by pathology. Distant metastasis was defined as abnormal changes or typical metastatic lesions in the internal organs or brain, combined with hematological examination of the tumor and was confirmed by radiological imaging. Most cases were also confirmed by biopsy. OS was defined as BC-related death.

Pathological sections were independently evaluated by pathologists at our hospital. The evaluation criteria for ER, PR, HER-2 used the guidelines for breast HR, PR, and HER-2 detection developed by ASCO/CAP (*Chen et al., 2022*). ER and PR positivity were defined as cells with $\geq$ 1% of nuclear receptor positivity. HER2 positivity was defined as a 3+ score by immunohistochemistry or gene amplification by fish. Ki67 negative tissues had a brownish yellow stain with <10% of cells stained positive, positive cells had >10% (+), 10–25% (++) or 26–50% (+++) positive staining. A high expression of Ki67 was defined as the average ratio of positive cells to all tumor cells $\geq$ 14%, and low expression was defined as <14%. bpCR was defined as the primary focus of the breast tumor without any residual disease. apCR was defined as no metastasis in the lymph nodes after axillary dissection and ipCR was defined as the absence of metastasis in lymph nodes after modified extended radical mastectomy or IMLN dissection (5).

The clinical staging of BC patients was performed according to the American Joint Committee on Cancer (AJCC) TNM staging system as follows; (1) tumor size (T), growth pattern (whether infiltrating Tis), (2) lymph nodes status (N) with or without metastasis including the size, number and distribution of metastases, (3) the distant metastasis of the tumor (M). Stage 0 = T N0M0; Stage I = T1N0M0; Stage II = T0N1M0, T1N1M0,

T2N0M0, T2N1M0, T3N0M0; Stage III = T0N2M0, T1N2M0, T2N2M0, T3N1~2M0, T4N0~2M0, any TN3M0; Stage IV: any T, any NM1.

## Statistical analysis

Statistical analysis was performed using SPSS 21.0 (SPSS Inc., Chicago, IL, USA). Normally distributed data were expressed as the X ± s and a group $T$-test was used for comparisons between the two groups. The counting data were expressed as cases or rates and the two groups were compared using a $\chi 2$ test. The Kruskal Wallis rank sum test was used to compare multiple groups of grading data, those with statistical significance in univariate analysis were included in multivariate analysis. A logistic regression model was used for multivariate analysis and a receiver operating characteristic (ROC) curve was used to evaluate the predictive value of relevant factors for ipCR. The Youden index (sensitivity + specificity) was used to calculate the optimal cut-off threshold value, and the corresponding AUC and 95% confidence interval (CI) were used to determine the specificity (SPE) and sensitivity (SEN). The Kaplan–Meier method was used for the analysis of OS and DFS.

# RESULTS

## Clinical features

Sixty-five female patients were included in the analysis. The patients had a median age of 46.5 years, seven patients were <35 years old (10.77%) and 58 patients (65.7%) were premenopausal. Five patients were stage cT1 (7.69%), 42 patients were stage cT2 (64.62%), nine patients were stage cT3 (13.85%), and nine patients were stage cT4 (13.85%). After neoadjuvant chemotherapy, 13 cases (20.0%) obtained a bpCR, 21 cases (32.31%) obtained an apCR, and 29 cases (44.62%) obtained an ipCR. 24 cases (36.92%) had recurrence or metastasis and 20 cases (30.77%) died after recurrence and metastasis.

## Single factor analysis of parameters influencing ipCR

The HR negative cases and APCR rates were shown to be significantly different between the ipCR group and non-ipCR groups ($P < 0.05$, see Table 1).

## Multiple linear regression analysis of the factors influencing ipCR

Statistically significant variables identified through univariate analysis were taken as independent variables and ipCR was used as a dependent variable (No = 0, Yes = 1) for multivariate logistic regression analysis. The results showed that HR (OR = 2.698) and apCR (OR = 4.546) were factors that significantly influenced ipCR ($P < 0.05$, see Table 2).

## The predictive value of HR and apCR in ipCR

HR and apCR were used as test variables and ipCR was the dependent variable (No = 0, Yes = 1). These data were used to generate a ROC curve. The AUC for ipCR predicted by HR was 0.744 and by apCR was 0.735. The AUC for joint detection was 0.905 (see Table 3 and Fig. 1 for details).

## Prognostic analysis

The average DFS of patients in the ipCR group was 94.0 months which was significantly higher compared to patients in the non-ipCR group (64.2 months) ($\chi 2 = 4.265$, $P = 0.039$,

**Table 1  Single factor analysis of factors affecting ipCR (examples (%)).**

| Characteristics | n | non-ipCR group ($n = 36$) | ipCR group ($n = 29$) | $\chi 2$ | P |
|---|---|---|---|---|---|
| Age (years) | | | | | |
| <35 | 7 | 2 | 5 | 2.282 | 0.131 |
| ≥35 | 58 | 34 | 24 | | |
| Menopausal status | | | | | |
| Pre- | 42 | 26 | 16 | 2.042 | 0.153 |
| Post- | 23 | 10 | 13 | | |
| T status | | | | | |
| T1 | 5 | 3 | 2 | 2.362 | 0.501 |
| T2 | 42 | 21 | 21 | | |
| T3 | 9 | 7 | 2 | | |
| T4 | 9 | 5 | 4 | | |
| HR | | | | | |
| Negative | 29 | 12 | 17 | 4.156 | 0.042 |
| Positive | 36 | 24 | 12 | | |
| HER2 | | | | | |
| Negative | 44 | 28 | 16 | 3.753 | 0.053 |
| Positive | 21 | 8 | 13 | | |
| Ki67 | | | | | |
| Low expression | 11 | 9 | 2 | 3.744 | 0.053 |
| High expression | 54 | 27 | 27 | | |
| Chemotherapy cycle | | | | | |
| ≤4 | 27 | 17 | 10 | 1.073 | 0.300 |
| >4 | 38 | 19 | 19 | | |
| Location | | | | | |
| Central | 23 | 13 | 10 | 0.256 | 0.880 |
| Inner | 16 | 8 | 8 | | |
| Outer | 26 | 15 | 11 | | |
| Lesion | | | | | |
| Single | 47 | 24 | 23 | 1.282 | 0.258 |
| Multiple | 18 | 12 | 6 | | |
| bpCR | | | | | |
| No | 53 | 32 | 21 | 2.896 | 0.089 |
| Yes | 12 | 4 | 8 | | |
| apCR | | | | | |
| No | 43 | 30 | 13 | 10.641 | 0.001 |
| Yes | 22 | 6 | 16 | | |

**Table 2  Multiple linear regression analysis of the influencing factors of ipCR.**

| Risk factors | $\beta$ | SE | Ward | OR | 95% CI | P |
|---|---|---|---|---|---|---|
| HR (No, for reference) | 0.992 | 0.426 | 5.428 | 2.698 | 1.170~6.218 | <0.001 |
| apCR (No, for reference) | 1.514 | 0.517 | 8.578 | 4.546 | 1.650~12.523 | <0.001 |

**Table 3  Analysis of the predictive value of HR and apCR on ipCR.**

| Index | AUC | 95% CI | Specificity | Sensitivity |
|---|---|---|---|---|
| HR | 0.744 | 0.604~0.883 | 70.14 | 69.87 |
| apCR | 0.735 | 0.593~0.877 | 71.87 | 70.54 |
| Joint testing | 0.905 | 0.823~0.987 | 85.17 | 81.76 |

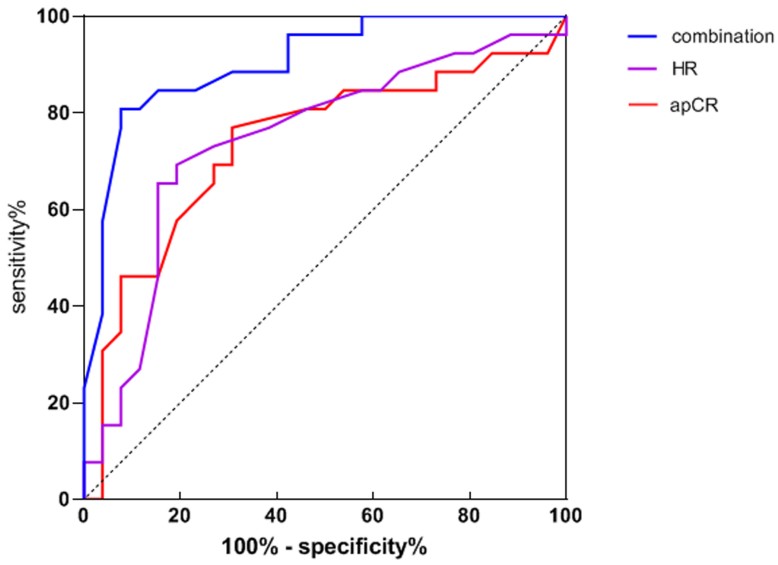

**Figure 1  Receiver operating characteristic of HR, apCR and joint detection to predict ipCR.**

see Fig. 2). No significant difference in OS was detected between the two groups ($p > 0.05$, see Fig. 3).

## DISCUSSION

Lymphatic drainage in the breast mainly occurs through the axillary and internal mammary lymph nodes. The ipsilateral ALN drains 75% of the lymph of the breast, while the remaining 25% of the lymph is drained from the IMLNs. Although the IMLNs are defined as regional nodes similar to the ALNs, previous studies have shown that BC cases with IMLN metastasis have a worse prognosis compared to cases involving metastasis to the ALNs (*Madsen et al., 2015*). *Noguchi et al. (1991)* retrospectively analyzed 286 BC patients who underwent IMLN dissection from 1956 to 1987. The study found that independent of ALN status, the 20-year DFS of patients with ILNM was significantly reduced (25% *vs* 67%; $p < 0.0001$). Only 44% of the patients received chemotherapy, 16% received endocrine therapy, and 5% received radiotherapy.

Clinical studies have highlighted that lymph node metastasis in the internal mammary region is a poor prognostic indicator independent of ALN status (*Karanetz et al., 2021*). However, local treatment of the IMLNs remains controversial. Randomized trials comparing extended modified radical mastectomy with modified radical mastectomy

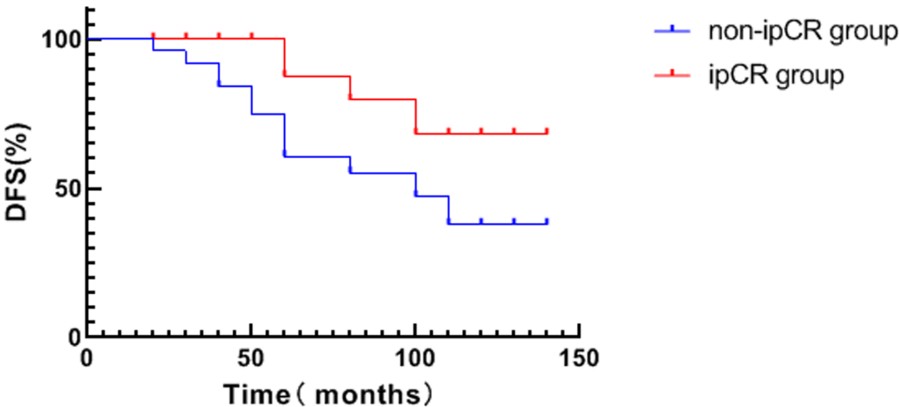

**Figure 2** Analysis of survival curve of ipCR and DFS in breast cancer patients.

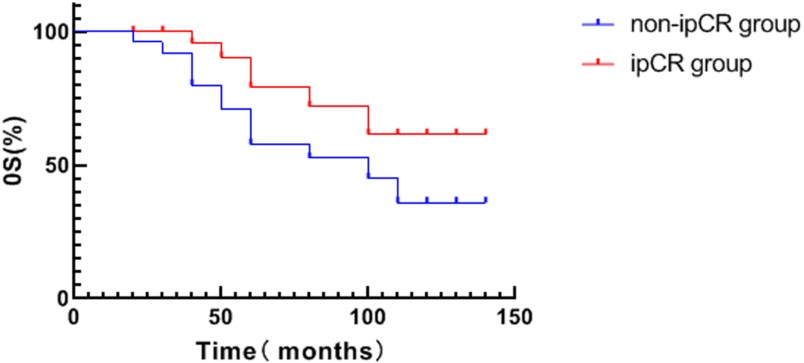

**Figure 3** Analysis of survival curve of ipCR and OS in breast cancer patients.

failed to show a significant benefit in survival. However, most of the early trials did not involve adjuvant treatment and some concerns remain. Recent results from randomized trials in Denmark and British Columbia and the EBCTCG meta-analysis suggest the importance of local regional control for survival (*Chen et al., 2008*). With the widespread application of systemic therapy, it remains to be considered whether the regional treatment of the endo mammary lymph outcome can translate into a survival benefit. At present, local control combined with systemic therapy is recommended, but the related factors affecting ipCR after NAC and the relationship between ipCR and prognosis are still unclear.

Neoadjuvant therapy is the standard therapy for locally advanced BC. NAC reduces the need for partial mastectomy, axillary dissection and the rate of surgery but does not increase the risk of local regional recurrence (*King & Morrow, 2015*). There are a few reports on the pCR of the IMLNs after neoadjuvant chemotherapy. This study found that the incidence of ipCR after NAC was 44.62%. This was significantly higher than compared to the rates of bpCR and apCR suggesting that lymph node metastasis in the inner mammary region can be cleared more easily by NAC.

Molecular markers play an important role in informing clinical decision making in the treatment of BC and can also play a key role in prognosis and evaluating response to treatment. The utility of molecular markers in predicting ipCR is yet to be reported. This study showed that patients with HR negative and APCR are more likely to obtain an ipCR which is consistent with results from other studies (*Jung et al., 2019*; *Johnston et al., 2023*; *Schipper et al., 2021*). Currently, resection of the IMLN remains controversial in clinical practice. Therefore, if the ipCR can be predicted by HR negativity and apCR, the necessity of IMLN dissection after NAC is also debatable.

According to the ROC curve in this study, HR and apCR for ipCR prediction were 0.744 and 0.735, respectively and the AUC of HR was 0.744. These data indicate that HR has predictive in ipCR after NAC. The probability of ipCR after NAC can be predicted to a certain extent by HR. However, the lower AUC value also means that the predictive ability of HR as a single predictor is relatively weak. The AUC value of apCR is 0.735. These data indicate that when apCR is used as a separate predictor, it also has predictive power in ipCR after NAC. However, the predictive power of apCR was lower compared to HR status. The AUC for combined detection was 0.905 and the area under the line was higher than for single detection. The AUC value of combined detection was 0.905. These data indicate that the combination of HR and apCR could significantly improve the predictive power of ipCR after NAC. The higher AUC value means that the combined detection had higher predictive accuracy for patients showing an ipCR.

In clinical practice, assessing the probability of ipCR after NAC is important for treatment decision-making. HR and apCR can provide predictive power to inform treatment decisions, prognosis and assess treatment responses. The combined detection of HR and apCR can further improve the accuracy of predicting ipCR after NAC and provides a reference for doctors to develop more accurate treatment strategies.

pCR is regarded as pathological evidence of the effectiveness of neoadjuvant therapy and is a key measure used to evaluate the efficacy of neoadjuvant therapy. pCR after neoadjuvant is a strong prognostic factor (*Cortazar et al., 2014*). Lymph node metastasis in the inner mammary region and ALN metastasis have similar prognostic importance. Whilst the traditional definition of pCR for lymph nodes assesses the pathological status of the breast and axilla, it ignores the pathological status of lymph nodes in the inner mammary region (*Bi et al., 2018*). The results of this study show that the patients with ipCR had a better prognosis. Kaplan-Meier survival analysis showed that the DFS of patients with ipCR was significantly higher than in the non-ipCR group ($P < 0.05$). These data suggest that ipCR is an important factor that affects the prognosis of patients with lymph node metastasis in the internal mammary region. Similar to bpCR and apCR, ipCR can better distinguish between patients with a good and a poor prognosis.

Despite the interesting data reported in this study, our data are subject to several caveats. This analysis was performed as a single-center retrospective study and had a relatively small sample size. More detailed prospective studies should be performed to include additional clinical data such as blood flow signals detected by ultrasound, the shape of the mass, inflammatory biomarkers and side effects such as postoperative complications and

lymphedema. This is the first study to evaluate the factors influencing ipCR and prognosis of ipCR in BC including pathological analysis of the lymph nodes.

Further studies are needed to comprehensively evaluate the status of the axillary and internal mammary lymph nodes and improve the definition of pCR of the lymph nodes towards improving treatment and prognosis.

## CONCLUSIONS

Whether to obtain ipCR after neoadjuvant chemotherapy is correlated with HR and apCR. HR combined with apCR has predictive power in ipCR and can be used to predict the prognosis of patients with internal mammary lymph node metastasis. Future research should focus on verifying the current findings, exploring new treatment strategies, and conducting larger-scale research to promote its clinical application.

### Funding
The authors received no funding for this work.

### Competing Interests
The authors declare there are no competing interests.

### Author Contributions
- Yang Li conceived and designed the experiments, analyzed the data, prepared figures and/or tables, and approved the final draft.
- Yang Fei performed the experiments, analyzed the data, authored or reviewed drafts of the article, and approved the final draft.

### Human Ethics
The following information was supplied relating to ethical approvals (i.e., approving body and any reference numbers):

All samples obtained in this study were approved by the ethics committee of the The First Medical Center of Chinese PLA General Hospital and abided by the ethical guidelines of the Declaration of Helsinki.

### Data Availability
The raw data are available in the Supplementary File.

### Supplemental Information
Supplemental information for this article can be found online at http://dx.doi.org/10.7717/peerj.16141#supplemental-information.

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
