# Peer review of "Analysis of factors influencing the pathological complete remission (ipCR) in patients with internal mammary lymph node metastasis after neoadjuvant chemotherapy"

_PeerJ, doi:10.7717/peerj.16141_

## Round 0.1 · original submission · Minor Revisions

The identification of specific factors that influence ipCR in patients with internal mammary lymph node metastasis adds to the understanding of tumor behavior and response to neoadjuvant chemotherapy. This study's findings contribute to the scientific knowledge base and may inspire further research in this area.

Please respond and make appropriate revisions based on the reviewers' suggestions and my comments (below). This will greatly improve the quality of the manuscript.

My comments:
1. The title of this manuscript could be further improved. For instance: Analysis of factors Influencing ipCR in patients with internal mammary lymph node metastasis after neoadjuvant chemotherapy.
2. Abstract and Conclusion sections: The conclusions shown in these two sections should be strengthened by discussing the clinical implications of the findings and suggesting avenues for future research.
3. Abstract section: HR or apCR means what? All abbreviations must be explained in detail and accurately when they first appear in the paper.
4. [there was no non cancer]?
5. [The average DFS of ipCR group was 94.0 months, which was significantly better than that of non-ipCR group 64.2 months] should be revised to [The average DFS of the ipCR group was 94.0 months, which was significantly better than that of the non-ipCR group (64.2 months).].

Reviewer 1 ·

Basic reporting

The manuscript demonstrates clear and technically correct English, making it easily understandable. The introduction and background effectively contextualize the study within the broader field of knowledge, highlighting the significance of understanding the prognosis of breast cancer patients with internal mammary lymph node metastasis. The authors identify a knowledge gap by emphasizing the lack of studies evaluating the influencing factors and prognosis of internal mammary lymph node metastasis after neoadjuvant chemotherapy. This study fills that gap by comprehensively evaluating the influencing factors and prognosis of internal mammary pathological complete response (ipCR) in breast cancer patients.
In terms of technical standard, the manuscript provides a detailed description of the methodology, allowing for potential replication of the study. The availability and robustness of the underlying data are not explicitly discussed, which could be a limitation. However, the statistical soundness of the analysis is confirmed through Kaplan Meier survival analysis and the use of proper statistical tests to assess the significance of the findings. Control measures for bias and confounding variables are not extensively mentioned, which may impact the reliability of the results.
Overall, the manuscript presents a valuable contribution to the field by shedding light on the prognostic importance of ipCR in breast cancer patients with internal mammary lymph node metastasis. The study's methodology and results are adequately presented, although further clarification on data availability and control measures would strengthen the overall evaluation.

Experimental design

1. Provide a brief explanation of the inclusion criteria: why these specific criteria were chosen and how they relate to the research question. Likewise, briefly explain the reasoning behind the exclusion criteria and why these specific conditions were considered inappropriate for this study.
2. Provide a clear description of the neoadjuvant chemotherapy regimens used in the IPCR and non IPCR groups.
3. Describe in more detail and provide clarity on the methodology and procedure used for telephonic follow-up, including the frequency and duration of follow-up calls and the specific data collected during these conversations.
4. Discuss the clinical relevance and appropriateness of the defined endpoints for disease-free survival (DFS) and overall survival (OS) in relation to the research question. Provide a brief explanation of why these endpoints were chosen and their significance in assessing the effectiveness of neoadjuvant chemotherapy.
5. Provide a clear rationale for focusing on DFS and OS as prognostic measures. Explain why these endpoints were chosen and their relevance to the research question. Consider discussing other relevant endpoints that could provide a more comprehensive analysis of prognosis.

Validity of the findings

1. Briefly discuss the rationale for using adjuvant endocrine therapy specifically for hormone receptor-positive patients and how it may affect the outcome measures.
2. Describe the specific neoadjuvant chemotherapy regimens used, including the drugs, dosages, and frequency of administration. This will provide better clarity and should be referenced to avoid ambiguity.
3. Discuss potential limitations and biases of the study, such as selection bias, missing data, and confounding variables. Address any possible sources of bias that may have influenced the results. This will improve the overall scientific integrity and validity of the study.

Additional comments

1. The introduction should provide more context and background information on breast cancer, internal mammary lymph nodes, and their significance. This will help readers understand the importance and relevance of the topic.
2. There should be a clear research question or objective stated at the end of the introduction to guide the reader and create a sense of purpose for the study.
3. Provide a more detailed explanation of neoadjuvant chemotherapy, its benefits, and its relevance to breast cancer treatment.
4. Expand on the lack of previous research on the factor analysis of internal mammary lymph nodes' pathological complete response (ipCR) after neoadjuvant chemotherapy. Why is this knowledge gap important to address?

Reviewer 2 ·

Basic reporting

The manuscript is well-written with high technical accuracy terms clearly defined, providing excellent understanding and grounding for a reader. The introduction and background provide sufficient information on the impact of internal mammary lymph node metastasis on the prognosis of breast cancer, highlighting its clinical importance and the current lack of understanding of predictive factors and prognosis related to pathologic complete response (pCR) in the internal mammary lymph region (ipCR). This demonstrates the gap in knowledge that the study aims to fill. The study stands out for its robust technical approach. Comprehensive statistical analyses including univariate and multivariate analysis, the Kaplan-Meier method, Logistic regression, and the receiver operating characteristic (ROC) curve were aptly utilized. The author's methodology is explained in great detail that could potentially allow for replication in other studies, despite some deficiencies such as a small sample size and the retrospective design. These limitations are acknowledged transparently, further reinforcing the credibility of the report. Some areas for improvement would be the provision of the underlying data to confirm the robustness and control measures of the study.

Experimental design

1) Provide a brief rationale for selecting 65 patients for the study. Why is this sample size appropriate?
2) Clearly state the type and duration of radiotherapy received by all patients after surgery, as well as the rationale behind this treatment choice.
3) Explain whether there were any missing or incomplete data and if so, how they were handled during analysis. Mention any methods used for imputation or any limitations introduced by missing data.

Validity of the findings

1) Specify the criteria used to divide the patients into the IPCR and non IPCR groups.
2) Provide a reasoning or previous evidence for choosing taxanes or anthracyclines as part of the neoadjuvant chemotherapy regimen. Explain how these choices may impact the likelihood of achieving IPCR.
3) Clarify the methodology and specific criteria used to determine clinical features such as CT stage. Provide a reference or explanation for the classification system used to ensure consistency with other studies.
4) Describe the methodology for drawing the ROC curve, including the choices of cutoff values, sensitivity, specificity, and the interpretation of the AUC values. This will provide more context and transparency in assessing the predictive value of HR and apCR.
5) Clarify the interpretation of the AUC values for HR, apCR, and joint detection in predicting ipCR. Explain the clinical significance of these values and their impact on decision-making. This will help readers understand the utility of these variables in clinical practice.

Additional comments

1) The introduction should mention the current knowledge gaps or limitations in the field of internal mammary lymph node metastasis and the need for further research.
2) Clarify the significance of pathological complete response (pCR) as an independent prognostic factor and its relevance to the study's objective.
3) State the specific aims or hypotheses of the study to help readers understand the research's purpose.
4) Check for grammatical errors and consistency in language usage throughout the manuscript.

Reviewer 3 ·

Basic reporting

The manuscript articulates well the knowledge gap regarding ipCR's predictive factors and its impact on the prognosis. Subsequently, it very well explains how the presented study addresses this gap. The technical standards applied in analyzing and interpreting the data are of high quality, utilizing diverse statistical methods with due diligence. The methodology is replicated with detailed transparency and precision in the procedures. However, the author could have presented the data explicitly to increase the manuscript robustness and ensure that readers could independently verify the results. Also, given the retrospective study design and the relatively small sample size, the author should be cautious about potentially overgeneralizing the findings. Nonetheless, with its nuanced analyses and detailed methodology, the manuscript makes a considerable contribution toward understanding ipCR factors, offering vital insight for breast cancer treatment.

Experimental design

I. Briefly discuss the rationale for selecting a sample size of 65 patients, considering factors such as statistical power, feasibility, and previous studies in the field.
II. Clearly explain the criteria used to divide the patients into the IPCR and non IPCR groups. This could include details on the pathological conditions of the internal mammary region after the operation.
III. Elaborate on how the assessment of local recurrence and distant metastasis was conducted, including the specific diagnostic methods employed (imaging, biopsies, etc.) and the role of pathology confirmation in each case.

Validity of the findings

I. Describe the specific neoadjuvant chemotherapy regimens used in the IPCR and non IPCR groups, including the drugs, dosages, and frequency of administration. This will provide better clarity and should be referenced if possible.
II. Clarify the criteria used to define Er (Estrogen Receptor), PR (Progesterone Receptor), HR (Hormone Receptor), HER2 (Human Epidermal Growth Factor Receptor 2), Ki67, and Pcr (Pathological Complete Response). Provide appropriate references or citations for these definitions to ensure scientific accuracy and consistency.
III. Clearly define and explain the outcomes of interest, such as recurrence, metastasis, and non-cancer related death. Specify the criteria used to determine recurrence and metastasis to ensure consistency and accuracy.

Additional comments

I. The statement that breast cancer has surpassed lung cancer as the cancer with the highest incidence in the world needs to be supported by reliable and up-to-date statistics. If possible, provide the source of this information.
II. The sentence regarding the controversial indications for biopsy or resection of internal mammary lymph nodes needs more explanation. What are the different viewpoints and why is there disagreement?
III. The statement that multimodal treatment is advocated for breast cancer with internal mammary lymph node metastasis should be supported by evidence and references.
IV. Consider defining any technical terms or abbreviations that may be unfamiliar to readers.
V. Discussion: Clarify the significance of the previous studies that have shown worse prognosis for breast cancer with internal mammary lymph node metastasis compared to axillary lymph node metastasis. Provide specific information on the findings of these studies.

---

## Round 0.2 · accepted · Accept

I was satisfied with the responses and revisions made by the authors. The Reviewer's and my concerns have been well addressed. With the necessary revisions and improvements, the quality of this paper has been significantly improved. I believe that this revised manuscript is ready to be considered for publication in this journal.

Reviewer 1 ·

Basic reporting

No comment.

Experimental design

No comment.

Validity of the findings

No comment.

Additional comments

The authors have addressed all of my comments and have incorporated those in the revised manuscript. Hence, I recommend to accept it for publication.

Reviewer 2 ·

Basic reporting

ok.

Experimental design

ok.

Validity of the findings

ok.

Additional comments

The author's revised manuscript can be published.

Reviewer 3 ·

Basic reporting

The author has made good revisions to this article and carefully revised my opinions. I have no other review comments.

Experimental design

Good.

Validity of the findings

Good.

Additional comments

Good.